# Study of various machine learning approaches for Sentinel-2 derived bathymetry

**Andrzej Chybicki**[1]*, **Paweł Sosnowski**[1], **Marek Kulawiak**[1], **Tomasz Bieliński**[1], **Waldemar Korlub**[2], **Zbigniew Łubniewski**[1], **Magdalena Kempa**[3], **Jarosław Parzuchowski**[4]

**1** Department of Geoinformatics, Faculty of Electronics, Telecommunications and Informatics, Gdansk University of Technology, Gdansk, Poland, **2** Inero Software sp. z o. o., Gdańsk, Poland, **3** Norwegian Institute for Water Research (NIVA), Oslo, Norway, **4** Interizon Cluster, Gdańsk, Poland

* andrzej.chybicki@eti.pg.edu.pl

**Data Availability Statement:** We published codes of the experiment results and all the data we used. In particular, we published bathymetry-estimator code on GitHub: https://github.com/coast-mapper/

## Abstract

In recent years precise and up-to-date information regarding seabed depth has become more and more important for companies and institutions that operate on coastlines. While direct, in-situ measurements are performed regularly, they are expensive, time-consuming and impractical to be performed in short time intervals. At the same time, an ever-increasing amount of satellite imaging data becomes available. With these images, it became possible to develop bathymetry estimation algorithms that can predict seabed depth and utilize them systematically. Since there are a number of theoretical approaches, physical models, and empirical techniques to use satellite observations in order to estimate depth in the coastal zone, the presented article compares the performance and precision of the most common one to modern machine learning algorithms. More specifically, the models based on shallow neural networks, decision trees and Random Forest algorithms have been proposed, investigated and confronted with the performance of pure analytical models. The particular proposed machine learning models differ also in a set of satellite data bands used as an input as well as in applying or not geographical weighting in the learning process. The obtained results point towards the best performance of the regression tree algorithm that incorporated as inputs information about data localization, raw reflectance data from four satellite data bands and a quotient of logarithms of B2 and B3 bands. The study for the paper was performed in relatively optically difficult and spatially variant conditions of the south Baltic coastline starting at Szczecin, Poland on the west (53˚26'17'' N, 14˚32'32'' E) to Hel peninsula (54˚43'04,3774'' N 18˚37'56,9175'' E). The reference bathymetry data was acquired from Polish Marine Administration. It was obtained through profile probing with single-beam sonar or direct in-situ probing.

## Introduction

Recent years have shown significant progress in the field of combining sonar and radar imaging for environmental monitoring. Current and stable growth in the field of sonar imaging technologies led to erecting worldwide available hydrographic data archives that can be used

bathymetry-estimator and on Zenodo: https://zenodo.org/record/6779671 This code can be run with the dataset, which is also published on Zenodo: https://zenodo.org/record/6543997 These resources are now also publicly available and have official DOI assigned: 10.5281/zenodo.6779671 and DOI: 10.5281/zenodo.6543997 respectively. Instruction on how to run computations with the mentioned dataset can be found in Readme file on GitHub (and inside zip file from Zenodo): https://github.com/coast-mapper/bathymetry-estimator/blob/main/README.md At the end of the Readme file, Manuscript Reviewers and potential Readers will find commands, which allow them to reproduce scientific experiments as performed in the paper. Near every command in square brackets is the model's name. The same names were used in the study.

**Funding:** The work performed in the study was co-founded by European Union Funds and Polish Government Funds under research works procured in the following agreements: 1. National Centre for Research and Development, Poland, Norwegian-Polish Cooperation Programme - POLNOR, grant no.: NOR/POLNOR/MPSS/0037/2019-00 2. Polish Agency of Development, Agreement no.: POPW.01.01.02-28-0038/21-00 3. National Centre for Research and Development, Poland, grant no.: WG-POPC.03.03.00-00-0007/17-00.

**Competing interests:** The authors have declared that no competing interests exist.

for various purposes. One of the interesting applications where sonar imaging data can be indirectly utilized for environmental monitoring is satellite-derived bathymetry (SDB).

SDB is an interesting alternative for sonar seabed imaging because it does not require high equipment investment such as survey vessels or hydroacoustic equipment. However, past sonar images are often used for calibration and verification of various SDB models and approaches. SDB is a field of research focused on the analysis of recorded levels of electromagnetic wavelengths dispersed over a location on Earth's surface. The general methodology of SDB is oriented to find the best possible match to estimate the actual seabed depth, taking into consideration various technical and physical phenomena and performing data analysis. Typically, EM radiation measurements are mostly acquired with dedicated probes installed on satellites i.a. Sentinel 2, Landsat 8, SPOT, and Pleiades. Between the commonly used wavelength ranges, one can identify the visible blue, red, green and near-infrared spectra.

This paper focuses on the comparison of several approaches that use statistical machine learning and compares them in the field of calculated root-mean-square error, coefficient of determination ($R^2$), and error distribution. Machine learning approaches are also compared to already studied methods based on polynomial regression algorithms [1] also known as inverse methods. The proposed models of machine learning are based on artificial neural networks as well as on regression trees and random forest algorithms. Each approach is presented with its respective learning methodology. The particular applied models differ also in a set of satellite data bands their ratios etc. used as an input, in applying or not geographical weighting in the learning process and in performing some additional operations like transforming the data to an exponential scale. The comparison focuses on the models' performance and the obtained bathymetry estimation accuracy.

In the next subsection, a brief introduction to satellite-derived bathymetry (SDB) is given along with a short review of related papers, followed by an introduction of the geographically weighted regression approach and its sample applications, while in the last subsection of this introduction the objectives of the presented work are stated.

## Introduction to SDB and GWR

In general, the sea bed shape estimation in Satellite Delivered Bathymetry (SDB) models relies on the assessment of the level of electromagnetic radiation dispersion in the visible (or near visible) spectrum. Based on that level and a specific empirical model, the depth of the sea bed is being calculated for a pixel corresponding to a particular geographic location.

SDB is a survey method that uses satellite or other remote multispectral imagery for depth determination [2] and has become a useful reconnaissance and planning tool for sea and ocean mapping. Especially, the usage of SDB is extremely practical in areas where vessel traffic or the deployment of aids to navigation indicates that charted data is misleading and there are no recent surveys to update the chart [3]. The results obtained from the SDB method could be applied in many hydrographic branches as well as in the marine sector, such as bathymetry, cartography, integrated coastal management, water coastal management, and water quality monitoring.

Although SDB provides bathymetry products at a coarser spatial resolution compared to traditional acoustic surveying or airborne lidar bathymetry (ALB), satellite imagery provides a routine repetitive coverage over the same area with advantages for the study of changes in dynamic seabed areas or areas with little survey data thus it can be also applied for chart adequacy assessment and survey prioritization [3]. It should be mentioned that SDB includes a variety of analytical, semi-analytical, and empirical methods. The first ones, analytical and semi-analytical SDB methods (called inversion models), were built upon the physics of light attenuation and thus require the parameterization of many optical properties of the

atmosphere, water surface, water column, and water bottom to derive water depth [4], under the assumption that these properties can vary spatially [5]. Consequently, these methods can generate more accurate estimations of water depth yet at the cost of model complexity and greater computational demand.

As an alternative, empirical methods estimate bathymetry based on statistical regression of depth (derived from in situ depth data collected at locations distributed throughout the study area) on the observed radiance of multiple light-penetrating bands, under the assumption of homogenous water columns (both vertically and spatially) and uniform bottom type across the image [6–8], the latter of which can be compensated through the use of multiband data.

One of the most commonly used algorithms is the log-ratio method developed by Stumpf, Holderied, and Sinclair [9], which uses the ratio of two log-transformed bands, typically the blue and green bands. This method assumes that both bands are similarly affected by the bottom reflectance and thus the ratio of them is insignificantly affected by the bottom type compared to water depth, causing the model to be partially independent of the bottom composition and benthic habitat.

On top of the dispersion and the analytical model, extra factors are taken into account. These might have influence over the satellite measurement of the energy level at a given band. Most common of these factors include technical factors such as sensor's characteristics and calibration parameters, orbit parameters, spatial resolution and saturation energy. Spatially correlated parameters such as seabed type, atmospheric correction, water turbidity, waving and other environmental factors are also taken into account, especially for locally geographically calibrated models.

Recently, the approaches based on machine learning algorithms, like random forest or different architectures of neural networks, have been introduced to SDB. In the reality, the relation between the electromagnetic radiance registered by satellite sensors and the water depth, utilized in SDB, is influenced by a large number of factors that complicates the physical description of the phenomena, and therefore modeling it by a simple formula may be not efficient. The use of machine learning algorithms for modeling more complicated forms of this relation could be the solution for this problem. One of the first works related to this issue was [10] where the multilayered perceptron network was used for coastal zone depth evaluation from airborne visible/infrared imaging spectrometer measurements. More recently, the random forest machine learning results were presented for SDB with the utilization of several channels: coastal, blue, green, red and near-infrared channels of Landsat 8 imagery along with a large amount of ground truth data [8]. Nowadays, deep learning methods along with several deep neural network architectures: convolutional, encoded-decoder, recurrent, and combinations of them, have been also introduced to SDB [11–13]. In [13], the indirect method of SDB relies on depth estimation from sea wave characteristics derived from a satellite image rather than from a pixel-based radiation level. From the most recent works, in [14] it was presented the application of several machine learning methods for SDB using very high-resolution multi-spectral Worldview-2 images along with the application of water quality inherent optical parameters obtained using the so-called updated quasi-analytical algorithm, showing that incorporating the latter improves the results. In another work [15], traditional SBD methods based on simple band ratios and their modifications were compared with those based on machine learning, namely, on Random Forest and XGBoost model, with obtaining markedly more accurate results for machine learning methods. In [16] the concept related to GWR was utilized along with machine learning for deriving bathymetry from optical images with a so-called localized neural network algorithm. The extensive and detailed up-to-date review of SBD methods and algorithms, describing and comparing a number of approaches, i.e. analytical, semi-analytical, statistical and empirical, including also those based on machine learning,

with finalization of the paper by providing a matrix as an aid for SDB data and technique selection for a needed level of accuracy, is given in [17].

The use of local spatially correlated factors for SDB was first introduced as geographically weighted regression (GWR) in 1996 in the geographical literature drawing from statistical approaches for curve-fitting and smoothing applications. This method works based on the simple, yet powerful idea of estimating local models, using subsets of observations centered on a focal point. Since its introduction, GWR rapidly captured the attention of many in geography and other fields for its potential to investigate nonstationary relations in regression analysis. The basic concepts have also been used to obtain local descriptive statistics and other models such as Poisson regression and probity. The method has been instrumental in highlighting the existence of potentially complex spatial relationships.

Today, GWR is a useful tool for inferring spatial processes, and a relatively simple and effective tool for spatial interpolation [18]. It is also a well-known powerful spatialization approach that enables flexible integration of predictors for systematic analyses of spatially varying relationships [19]. GWR became an established standard GIS routine and presently is applied widely in climate spatialization (e.g., [20–22]).

Although GWR allows for a spatially differentiated analysis of predictor-predictand relationships, one has to keep in mind that these are statistical relations and their physical meaning requires careful examination. Particularly when analyzing, for example, elevation-temperature dependencies in areas of rather low relief energy, GWR may yield physically implausible lapse rates which, in the case of sparse data coverage in adjacent mountainous areas, would serve to bias the spatialization results.

[22] was among the first works where GWR was applied in remote sensing. The technique was demonstrated by examining the normalized difference vegetation index (NDVI) for the rainfall relationship. Some recent examples include modelling the net primary production of Chinese forest ecosystems [23], the estimation of leaf area index (LAI) over a tropical rainforest [24], as well as tree diameter modelling using airborne lidar [25].

For the need of the marine environment, GWR is a reliable tool for estimating the natural variation of salinity in shallow seas. Such a system was used for the southern part of the Baltic Sea as well as for Florida Bay. GWR could be very useful in salinity modelling, because salinity may have strong local variations. It could be rather useful for shallow waters because they have a higher degree of optical complexity than clear deep oceanic waters [26].

GWR is designed to incorporate spatial dependency (non-stationarity) into regression models. To achieve this goal, a local regression model is constructed at every location and different relationships (regression coefficients) are possible at different points [21, 27]. GWR is therefore a truly local approach capable of capturing nonuniform spatial dependencies [28].

Recent GWR approaches are more frequently based on empirical techniques. Frequently, like in SDB and in many other issues of remote sensing at present, the machine learning models are used there to find optimal performance. Usually, different types of machine learning algorithms, e.g. neural networks of several architectures, XGBoost, random forest, ensembles of neural networks, or other learners, are combined with the use of geographically defined weights in the applied learning cost function. The description of this concept may be found for instance in [29]. An example of this approach utilizing machine learning and GWR in SDB is given in [16]. According to other topics, geographically weighted machine learning has been applied, for instance, in forest biomass estimation from satellite imagery [30], in high-resolution spatiotemporal estimations of wind speed from various environmental data [31], and in an investigation of hard coal mining impact on the natural environment [32]. In all these works, a simple, unified form of a radial geographically weighting function is used, depending only on the geographic distance between two points (the example may be the function given

by (5) in the next section). It was noticed in [33] that sometimes the geographical properties and spatial dependences describing given phenomena may be more complex and the use of the above approach for modelling them might be insufficient. Therefore the dedicated component of the neural network architecture has been proposed in this work for modelling the fluctuations in GWR, the so-called geographically neural network weighted regression (GNNWR). This concept has been applied for the estimation of the influence of the red tide on several elements of the sea environment [33].

## The research objectives

As it has been shown above, there exists a multitude of techniques and algorithms used in SDB, including analytical, empirical and physical approaches. Currently, there is no one universal method that would be considerably better than the others for every use case (see, e.g. [17]). What is more, it is in general difficult to compare directly the results of methods applied in different studies, because, apart from the used method itself, the performance may depend strongly on various conditions, e.g on availability and quality of calibration data sets, quality of acquisition, meteorological conditions, or, in a case of machine learning, also on many details on data preprocessing or used hyperparameters of a given model, or even on the utilized particular software package or library.

In this context, due to there are no many papers focused on performance comparison, with respect to the same measurement dataset, of a number of SDB methods differing not only in used approach but also in many other conditions and details, the objectives of the presented work may be stated as follows: to compare, on the same dataset, the performance of several machine learning methods against polynomial regression algorithms used as the reference approach. The investigated particular SDB models are differing also with respect to several detailed settings, including:

- a particular set of satellite data bands, their ratios etc. used as an input,

- applying or not geographical weighting in the learning process,

- performing some additional preprocessing operations like transforming the data to exponential scale.

The comparison has been made in terms of root-mean-square error, correlation and error distribution.

## Materials and methods

### Definition of the problem in the context of machine learning

Classical approaches to SBD rely on the phenomenon of light passing through a water column of a certain depth described by the Beer-Lambert law:

$$I = I_0 e^{-kz}, \tag{1}$$

where $I$–light energy after passing the water column of depth (height) $z$ and light attenuation coefficient $k$, $I_0$ –light energy before passing the water column. From satellite remote sensing data, we can derive the reflectance for a shallow water area for more than one electromagnetic band, e.g. for visible blue, visible green and even more bands. However, due to strong variability of several factors influencing the radiance value registered by the satellite sensor, i.e. air-water interface reflectance, wave attenuation in the water column and water-seabed reflectance, it is not easy to obtain a simple, universal formula for efficient sea depth estimation from satellite imagery. On the other hand, usually we may have at our disposal a set of sea

depth data acquired by more accurate methods than satellite imaging, i.e. *in-situ* echosounder measurements. These data may be used for calibration of methods and models used in SDB.

In general, we may define the problem of SBD as an optimization of the function:

$$\hat{z}(x, y) = f(B_1(x, y), B_2(x, y), \ldots, B_{Nb}(x, y)) \tag{2}$$

where $\hat{z}$ – sea depth of a given location $(x, y)$ estimated by SDB, $B_1, B_2, \ldots, B_{Nb}$–reflectance (pixel) values acquired by satellite sensor for this location for a set of bands, $Nb$–number of used bands. The optimization is done using the calibration depth measurement data $z_1, z_2, \ldots, z_n$, $n$–number of the used calibration data points, what denotes the minimization of the cost function:

$$J = E\left[ \sum_{i=1}^{n} (z_i - \hat{z}_i)^2 \right], \tag{3}$$

where $E[\cdot]$ is an expected value operator, $n$–number of training (calibration) points, $z_i$–depth of $i$-th calibration point, $\hat{z}_i$ – depth of $i$-th point estimated by SDB. Due to expected geographical variability of the optimal form of $f(\cdot)$ function from (2) [1], the optimization may be performed locally instead of globally, assuming an adequately, geographically weighted influence of particular $z_i$ values on $\hat{z}_i$ estimation for a given location $(x, y)$.

Instead of assuming the closed, analytical form of a function from (2) and estimating, for instance, only some of its parameters, the dependence of depth $z$ on bands values $B_i$ may be derived by means of machine learning, e.g. using an artificial neural network with appropriately designed architecture, or other machine learning methods. This approach is applied in the presented work. The investigated machine learning models, accompanied by the reference not AI-based models implemented for results comparison, are presented in the following subsections.

## Proposed machine learning models for SBD

For the presented study, we have prepared 12 different machine learning models optimizing the $f(\cdot)$ from (2), described below. Each model was calibrated on the basis of the same data calibration set and observations. Works on the development of the algorithm for bathymetry estimation focused on multitude approaches, mostly based on shallow neural networks and regression trees that use localized geographical models for calibration. On top of that, four models which do not rely on AI were included for reference purposes.

All models were implemented in Python using numpy and scipy libraries (the latter was selected for its ability to deal with sparse matrices' calculations). Neural network local models were prepared using Keras and Tensorflow libraries. This allows for a flexible description of the algorithm as a graph operating on tensors. The graph itself contains parameters, which are being calculated during the algorithm's learning process. In the case of neural networks, the parameters represent the weights of specific neurons. The networks were finally composed to obtain a specific geographically weighted model (GWM) using Keras library.

The general model training schema is presented in Fig 1. Input data tuples are combined from Sentinel-2 MSI data and bathymetry reference data. Entire set is divided randomly to Test data (20%) and base training data (80%). If a given model supports validation data, base training data set is divided randomly into validation data (16%) and training data (64%). Training data and validation data (if supported) is used to train model. This process produces trained model and training results and metrics. Test data is used on trained model to perform final verification.

**Reference models.** The following models do not rely on AI and were included and implemented for reference purposes

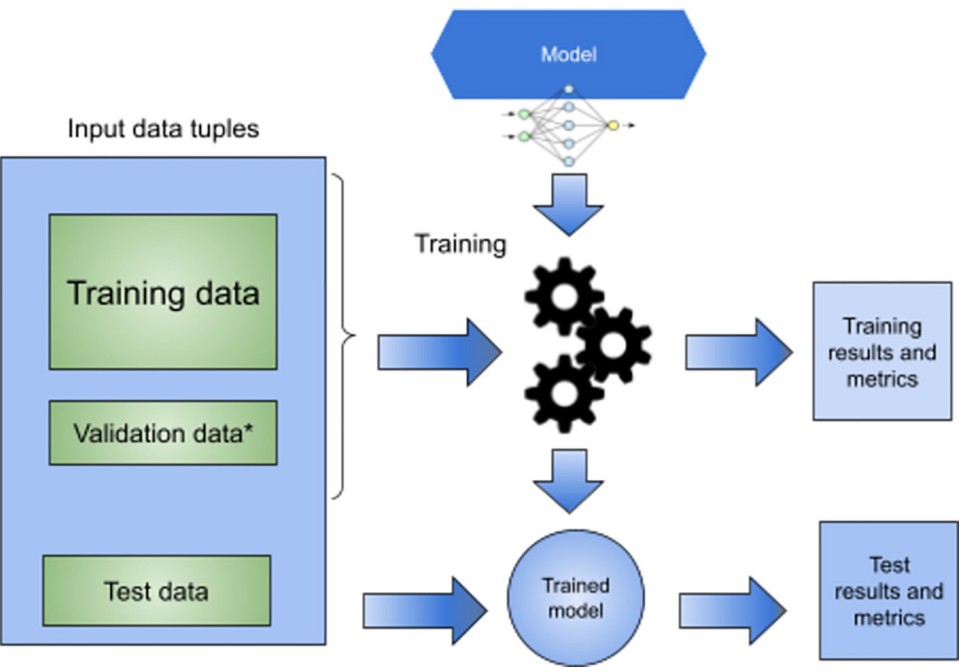

**Fig 1. General model training and testing scheme.**

**ref** This model implements the reference approach based on geographically weighted linear regression. It relies on the use of the ratio of logarithms of the visible blue band ($B_b$) and visible green band ($B_g$) reflectance for depth estimation at a given location $(x, y)$, originally proposed by Stumpf [9]:

$$\hat{z}(x, y) = {}_0 + {}_1 \frac{lnB_b(x, y)}{lnB_g(x, y)} = {}_0 + {}_1 \text{logratio} B_{23}, \tag{4}$$

where the 2-element vector of parameters $\alpha = [\alpha_0 \; \alpha_1]$ may be estimated by linear regression as shown in [1]. Further in the text the $B_b$ and $B_g$ bands are referred as $B_2$ and $B_3$ respectively, and their ratio is denoted as logratio$B_{23}$, as they correspond to the 2$^{\text{nd}}$ and 3$^{\text{rd}}$ spectral channels in the used Sentinel 2 imagery.

Due to not sufficiently good results of global optimization of $\alpha$ values caused by geographical variability of the conditions influencing satellite bathymetry, the GWR has been introduced to this approach. It relied on defining the local spatial weight of importance function $w_l(x, y)$ of the inverse distant form:

$$w_l(x, y) = \left[ 1 - \left( \frac{d_l}{b} \right)^2 \right]^2, \tag{5}$$

where $d_l$ is the distance of the point located at $(x, y)$ to the $l$-th local model window centre and $b$ is the geographical window width [1].

This approach has been implemented in Python. Weighted linear regression for matrix calculations is used to obtain the coefficients. Training of the models is performed on small, overlapping parts of the coast line, referred to as local data. The model was calibrated on an open sea area. Reference data comes from measurements obtained from Polish marine institutions.

Single input is prepared as a quotient of the logarithms of B2 and B3 reflectance bands logratio$B_{23}$ coming from Sentinel 2.

**exp** As in the ref model, it uses local GWM. It was calibrated on an open sea area. This approach takes into account and fits more to the exponential absorption of light radiation with water depth (height of the water column). It is based on the **ref** model with additional transforming the used logratio$B_{23}$ values to the exponential domain. It is achieved by applying the exp function after calculating the quotient of the logarithms of $B_2$ and $B_3$ reflectances. Similarly, the reference $z$ data values prior to utilizing them have a log function applied to them. During application of the model for seabed estimation, the obtained data passes through the exp function.

**gwmk** implementation of GWM algorithm using Keras. Opposite to the reference GWM implementation, the weighted linear regression is performed on the whole input data set (whole seacoast). The inputs are composed of quotients of logarithms of B2 and B3 resistance bands coming from Sentinel 2. Due to utilization of Keras library, this implementation differs from the original implementation of the **ref** model in several details, including the regularization method applied in the used linear regression calculation scheme.

**gwmk_ref** while the gwmk model used the whole data set as input, this approach returns to the original idea of creating local submodels. These are taught using small overlapping parts of the coastline. This model is the most similar to the **ref** model, differing only in the implementation details.

**Neural network models.** The following list presents the models based on artificial neural network.

**gwmk_nn** GMW model was implemented in Keras, where linear regression layers were replaced with shallow neural networks. In this approach, each neural network has the following architecture: one input neuron, 10 neurons with tanh activation function followed by 1 output neuron with a linear function. The model is being taught as a whole (without separation of input data to local subsections). The quotient of logarithms of B2 and B3 bands logratio$B_{23}$ is used as input. Here, the neural network optimizes the 1-variable function $f$(logratio$B_{23}$), the general form of which is dependent on details of the applied NN architecture.

**gwmk_nn_exp** the model is structured the same as gwm_nn, except for the output neurons of the layers' neural networks, where instead of a linear function, an exponent is applied.

**gwmk_nn_bands** Again, the model is structured similarly as gwmk_nn. This time the input was extended from a single neuron taking the quotient of logarithms to fully connected four neurons taking the reflectances of B2, B3, B4 and B8 bands.

**gwmk_nn_bands_qlog** The model works exactly as gwm_nn_bands, with the addition of a quotient of logarithms of bands B2 and B3 to the input, making 5 fully connected neurons at the first layer.

**Regression trees and random forest models.** On top of the neural network implementation, the approach based on regression trees (decision trees) and random forest algorithm was investigated. Python library sci-kit learn (sklearn) [34] was used for the implementation. In this case, the procedure of generating a regression tree relies on the splitting of input data by taking into account a single parameter at every node. Such splitting is repeated on the resulting subsets until the obtained set consists of a predefined size (for example, it is 0.1% of the input). Such a subset becomes a leaf of the tree. The result ascribed to it is the mean value of the data contained within it (in this case it is the average of the estimated seabed depth of the subset).

A significant and well-known issue of the decision tree approach utilized for regression purposes is the quantization of results. In trying to overcome this problem, a random forest approach was applied. Here, instead of using a single tree as a result generator, multiple ones were applied. Each tree is trained using a random subset of total input. The final result is the interpolation of sub-results of the used trees.

In the presented studies, the maximum depth of each tree was set to 100 (in most cases it oscillated around 15). The minimum data size required to split a node into 2 branches was 1% and the minimum node size which made it become a leaf was 0.1%. The remaining parameters for setting up the regression forest were taken from the default settings of sklearn library (*sklearn.tree.DecisionTreeRegressor* and *sklearn.ensemble.RandomForestRegressor*). For the random forest approach, a total of 100 trees was utilized.

The following list presents the models developed fot the purpose of the study:

**rt_qlog** A model that uses a regression tree which takes the quotient of logarithms of band data logratio$B_{23}$ as an input.

**rt_qlog_bands** A version of the above-mentioned model (**rt_qlog**), where the input is extended from only the quotient of logarithms or B2 and B3 reflectances to include raw reflectance data from all Sentinel 2 bands (i.e. B2, B3, B4 and B8).

**rt_spatial_bands** Another regression tree algorithm. It takes as input the reflectance data and UTM coordinates of the measurement. In such a manner, the results are taking into account the local characteristics of the data.

**rt_spatial_qlog** Regression tree algorithm that takes as input the logratio$B_{23}$ and UTM coordinates of the measurements.

**rt_spatial_qlog_bands** A mix of two previous models, where the decision tree is fed reflectance data, the logratio$B_{23}$ and UTM coordinates of the measurement.

**rf_bands** A model based on Random Forest algorithm, which utilizes the reflectance of all available bands for input.

**rf_qlog** This model is based on Random Forest algorithm which uses the quotient of logarithms of B2 and B3 bands as input.

**rf_qlog_bands** A combination of both previous models, with Random Forest as the main algorithm. It includes all available band's reflectances and the logratio$B_{23}$ as input.

In the context of satellite data processing, attention should also be paid to the aspect of the linear relationships between the values registered in individual spectral bands. This feature can cause inaccuracies and problems with estimating the correct baseline values when using machine learning tools. In practice, most approaches utilize either theoretical models describing the relationships between individual channels, or the correlation analysis between individual random variables is performed and its results are included in the final model. For this reason, in our case, as part of the comparative analysis, we decided to use the RF model, which performs relatively well with cases of input random variables collinearity. In our research work, we compared the results from the RF model to data from other models [35, 36].

**Training and reference data location.** The reference bathymetry data was acquired from Polish Marine Administration. It was obtained through profile probing with single-beam sonar or direct in-situ probing. The data range contains the entirety of Polish sea coast, starting at Szczecin on the west (53˚26'17" N, 14˚32'32" E) to Hel peninsula (54˚43'04,3774" N 18˚37'56,9175" E). However, training and validating SDB methods for the purpose of this study were limited to the northern side of the Hel Peninsula. Applied methods incorporated localized calibration or local modeling of optical conditions to calculate the bathymetry models. Local models were then included using weighted estimation to obtain global optimization for the entirety of the testing ground.

## Results and discussion

Comparison of acquired results is presented in Table 1. The names presented in column 2 correspond to the naming conventions presented in the previous sections. Column 3 stands for root mean square error (1), column 4 provides the correlation coefficient *R* (2) and column 5

**Table 1. Comparison of RMSE and coefficients of determination obtained using presented models.**

|  | Model name | RMSE [m] | $R^2$ |
|---|---|---|---|
| 1. | ref | 1.82 | 0.85 |
| 2. | exp | 2.92 | 0.75 |
| 3. | gwmk | 2.05 | 0.83 |
| 4. | gwmk_ref | 2.22 | 0.813 |
| 5. | gwmk_nn | 1.55 | 0.87 |
| 6. | gwmk_nn_bands | 0.96 | 0.92 |
| 7. | gwmk_nn_bands_qlog | 0.76 | 0.94 |
| 8. | gwmk_nn_exp | 1.73 | 0.85 |
| 9. | rt_bands | 1.03 | 0.91 |
| 10. | rt_qlog | 1.74 | 0.74 |
| 11. | rt_qlog_bands | 0.86 | 0.94 |
| 12. | rt_spatial_bands | 0.84 | 0.94 |
| 13. | rt_spatial_qlog | 0.76 | 0.95 |
| 14. | rt_spatial_qlog_bands | 0.69 | 0.96 |
| 15. | rf_bands | 1.04 | 0.91 |
| 16. | rf_qlog | 1.76 | 0.74 |
| 17. | rf_qlog_bands | 0.9 | 0.932 |

is the $R^2$ coefficient:

$$RMSE = \sqrt{\frac{\sum_{i=1}^{N}\left(\hat{z}_i - z_i\right)^2}{N}} \tag{6}$$

$$R = \frac{\sum_{i=1}^{N}(\hat{z}_i - \bar{\hat{z}})(z_i - \bar{z})}{\sqrt{\sum_{i=1}^{N}\left(\hat{z}_i - \bar{\hat{z}}\right)^2}\sqrt{\sum_{i=1}^{N}\left(z_i - \bar{z}\right)^2}} \tag{7}$$

where $\bar{\hat{z}}$ is the expected value (mean) of SDB—$\hat{z}_i$, $\bar{z}$ is the expected value (mean) of $z_i$ (depth obtained from calibration data) and $N$ is the number of elements in the tested population.

Scatter plots demonstrating each models' result precision regarding the test data are presented in Fig 2. Example visualization of bathymetry estimation using rt_spatial_qlog_bands algorithm is presented in Fig 3 (port of Gdynia, 54˚31'09" N 18˚32'22" E) and Fig 4 (Jastarnia, 54˚41'58" N 18˚40'36" E).

It may be seen from the results that the overall performance of the investigated algorithms is satisfactory and generally in line with already published studies. Generally, the RMSE, which is the main evaluation criteria for most SDB approaches, falls in the range between 0.75 m to 2 m for most cases. Furthermore, predominantly it falls below 1.5 m, thus the obtained RMSE is not worse than in cases presented e.g. in the detailed review in [16]. It may be noticed that in some studies, for instance, [14] or [16], the RMSE about 0.5 bas been achieved for most cases, but regarding what was mentioned above about the strong dependence of the method performance on various conditions of a given experiment, one should be careful in making categorical conclusions form direct comparisons of different studies results. For the same reason, further discussion will concentrate on the internal comparison of the results obtained in this study by application of different SDB models.

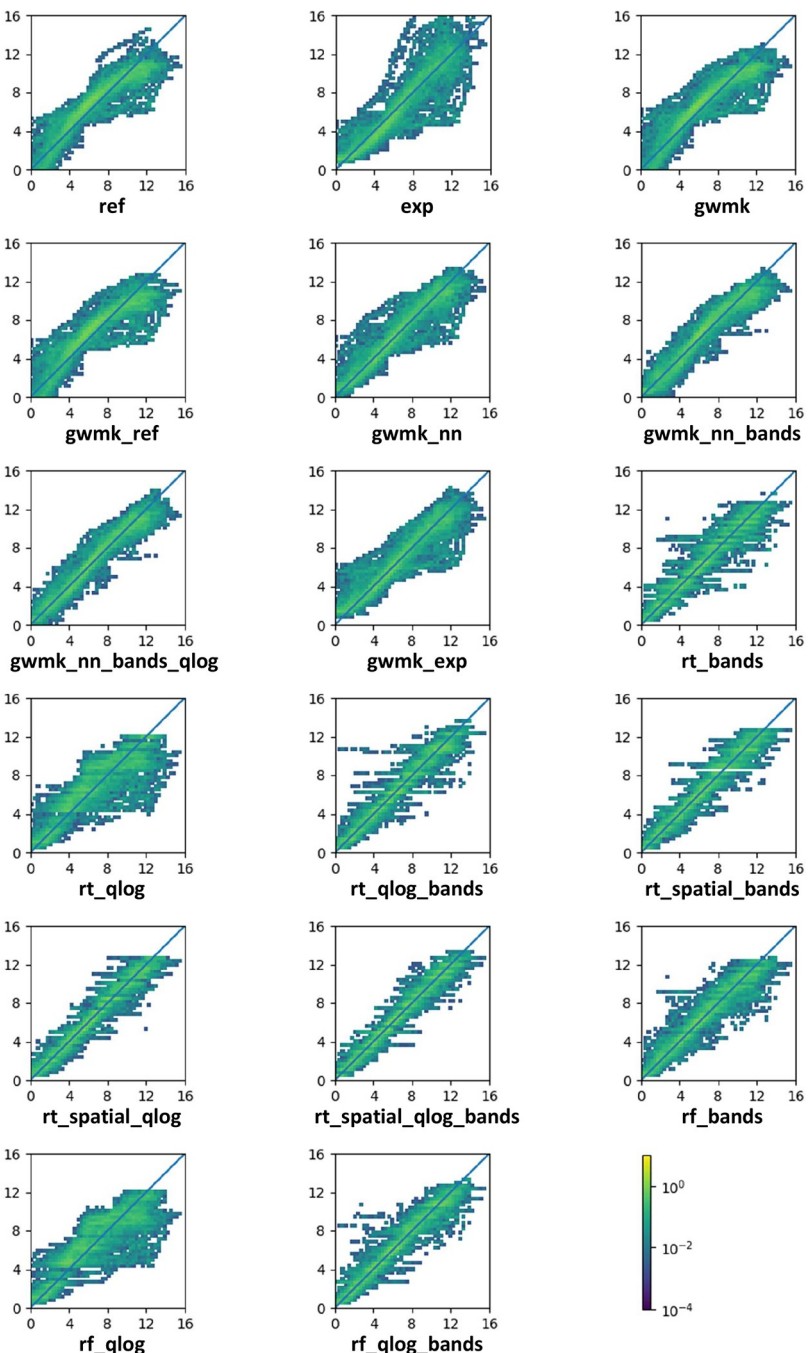

**Fig 2. Scatter plot of percentage of points that acquire certain precision for given test data.**

## RMSE

Below, the RMSE obtained for particular models (subsequent rows in Table 1) will be compared with that corresponding to the reference model ref.

**gwmk** and **gwmk_ref** As expected, the reimplementation of the well-known GWR algorithm in Keras (gwmk) did not improve the performance. On the contrary, the error increased by 12%. Even more discouragingly, applying localized data to Keras models as inputs in model gwmk_ref (as in the reference work) performed even worse, increasing the error by 22%.

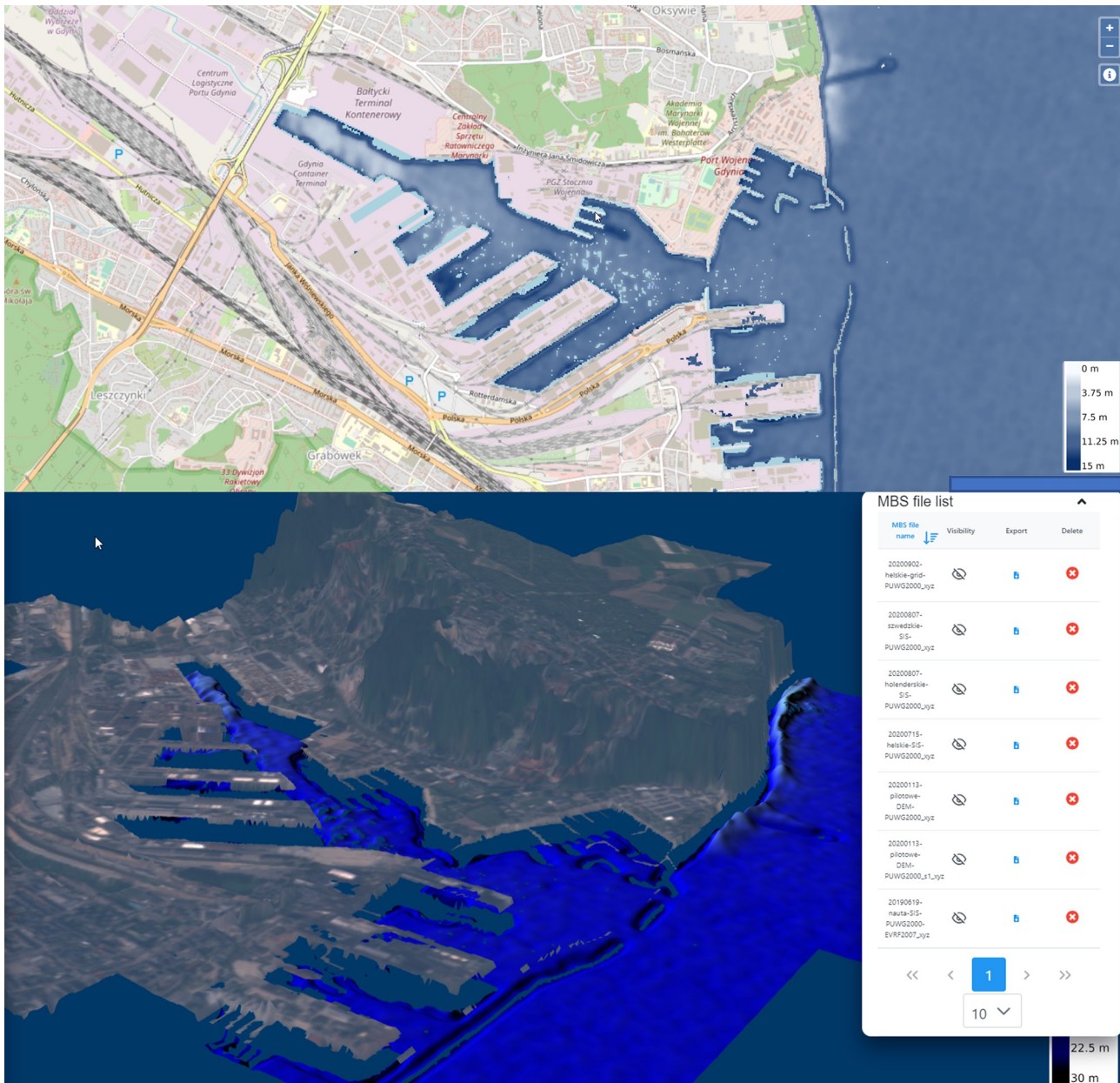

**Fig 3. Spatial 2D and 3D visualization of bathymetry estimations for Gdynia(geospatial source data: Open Street Map).**

The introduction of artificial intelligence was a step in the right direction. With the linear regression layers replaced by neural networks in the model gwnk_nn, the error was reduced by 17%.

The selection of specific data as inputs proved to be of great importance. While using inputs as per the original work of Chybicki [1] yields improved performance, it performs worse than inputs composed of direct band data from Sentinel 2 satellites in model gwmk_nn_bands with RMSE of 0.96.

Having that said, it is noted that when an approach incorporates together the direct band data and the quotient of bands' logarithms, like in gwmk_nn_bands_qlog, the error is reduced even further to 0.76 m.

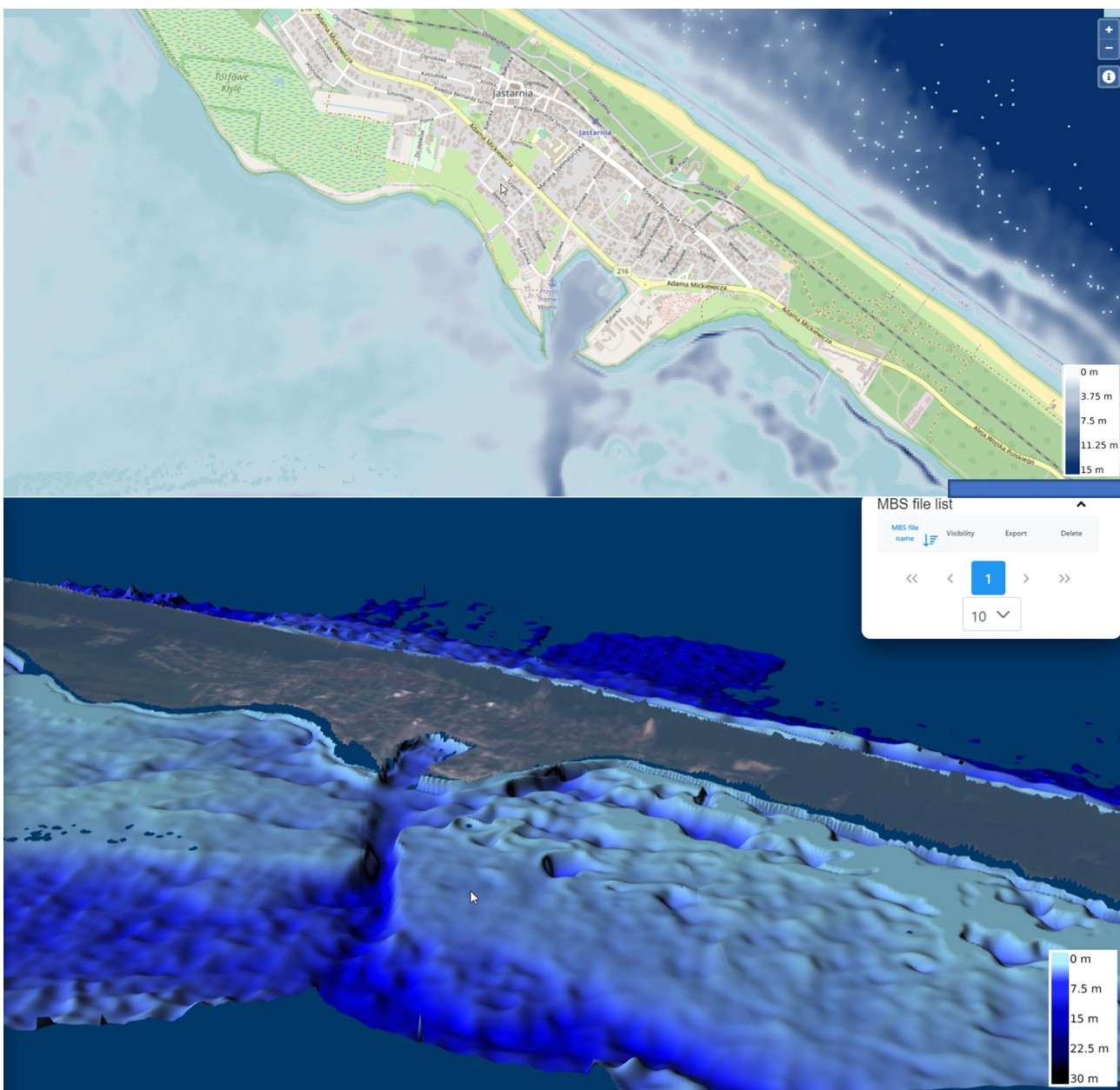

**Fig 4. Spatial 2D and 3D visualization of bathymetry estimations for Jastarnia (geospatial source data: Open Street Map).**

Using the exponent at the output neuron of the network, as in gwmk_nn_exp occurs as a suboptimal approach since the performance is worse than that of the algorithm with linear output.

At first, the introduction of regression tree algorithms seemed like not the best approach. An implementation which takes the quotient of logarithms as input performed worse than the analogical neural network approach of gwmk_nn, having an RMSE of 1.74 m versus 1.55 m of the latter. Similarly, approaches taking raw reflectance data and a mix of reflectance and quotient approach for regression tree algorithms (that is rt_bands and rt_qlog_bands) performed roughly 10% worse than their neural network counterparts. It is important to note that

regression tree algorithms do not inherently take into account the local character of the data, whereas neural network approaches do. Introducing information regarding input data localization to regression tree algorithms (as per rt_spatial_qlog, rt_spatial_bands, rt_spatial_qlog_-bands) significantly improves the performance, leading to the lower error for rt_spatial_qlog_bands approach with RMSE of 0.69 what is the best-obtained result according to RMSE.

**Random forest models** A natural step when dealing with the class of decision tree algorithms is to test the random forest approach. It was assumed that with the randomization of input data batches it would be possible to reduce the potential overlearning of the models. But unfortunately, the results of using random forest algorithms did not yield satisfactory improvements. For each input data class (quotient of logarithms, raw reflectances and the combination of both), the approaches performed slightly worse than their decision tree equivalents. Having in mind the extra complexity introduced by implementing random forest compared to regression tree algorithms, this approach was abandoned.

## Coefficient of determination and overall performance

At this point, it is important to address the overall performance of the studied algorithms regarding depth estimation on test data, with respect to the correlation between the reference and estimated depth and to obtained estimation accuracy for particular ranges of values (Fig 2). In addition to measures such as RMSE, it is also important to take into account the relative error in the analysis of the obtained results. However, it should be borne in mind that in the case of SDB analysis, such a comparison may not be reliable, because the error characteristics are strongly dependent on factors that are difficult to take into account in empirical modeling or measurement data such as high variability of the light scattering pattern from the waves for small depths or the temporal and spatial variability of bathymetry, especially apparent for the sandy bottom. For this reason, as part of the analysis and in addition to the analytical data, we also focused in the discussion of the results on the error characteristics depending on the depth and the algorithms used.

The performance of the original GWR model is presented in detail in Fig 2 (ref). It can be observed that for depths down to 8 m the model tends to overpredict the depth, while for larger depths an underprediction is noticed. The model which takes into account the exponential radiation absorption of water (Fig 2 exp) significantly improves the symmetry of the acquired results (i.e., it was not more overpredicting than underpredicting), although it maintains (or even increases) their spread.

Reimplementation of the GWM algorithm in Keras maintained the shape of the estimated data versus test data. It again demonstrated overprediction for smaller depths, and underprediction for larger ones, as shown in Fig 2 gwmk and Fig 2 gwmk_ref.

The introduction of neural networks slightly reduced the overestimation problem (see Fig 2 gwmk_nn and Fig 2 gwmk_nn_exp). At the same time, it did not result in the reduction of estimation spread. On that note, the inclusion of direct band data to the inputs did eliminate far-over- and far-underestimates (Fig 2 gwmk_nn_bands and Fig 2 gwmk_nn_bands_qlog).

A noticeable feature of the results provided by regression trees and random forest algorithms is the quantization of data. This is expected, as the result space for such algorithms is discrete. According to the performance comparison, as in the case of neural network algorithms, it was observed that using raw bathymetry as input data does not end with a satisfactory outcome (Fig 2 rt_qlog, Fig 2 rf_qlog). The estimations are overpredicted and are laden with big errors. Making use of EM reflectance data from all available bands removes this issue and significantly improves the performance of this class of algorithms (Fig 2 rt_bands, Fig 2

rf_bands) and when these inputs are combined with the raw bathymetry the outcome proves to perform even better (Fig 2 rt_qlog_bands, Fig 2 rf_qlog_bands).

Good performance is obtained by including data localization to the input data for regression tree algorithms. The best performing combination of input data proves to be the one that includes both reflectance data and raw bathymetry (compare Fig 2 rt_spatial_bands, Fig 2 rt_spatial_qlog and Fig 2 rt_spatial_qlog_bands). Not only does this significantly reduce the errors, but also provides correct estimations for the whole range of depths. The errors are still present but are significantly reduced and restricted to the quantified values generated by the regression tree.

Visualizations of bathymetry estimations for two representative example locations are presented in Figs 3 and 4. Results obtained for the port of Gdynia (Fig 3) visualize the local character of the training data sets. The transition of seabed depth estimation from the main port area to the northern local set is discontinuous, which is clearly an artifact resulted by overlearning of the algorithm in a specific area. Bathymetry data calculated for the harbor of Jastarnia (Fig 4) demonstrate the usefulness of satellite-delivered bathymetry methods for seabed modelling by clearly being able to identify the harbor's approach path. With satellite data being available significantly more frequently than in-situ measurements, numerical models provide an accurate approximation of changes in seabed shape in time, which may allow authorities to react in time in case of any unexpected disturbances.

## Conclusions

This article presents the comparison of multiple improvements to SDB algorithms that allow for the estimation of seabed depth using satellite images. In general, the obtained depth estimation performance is satisfactory with comparison with results reported in other studies. From the obtained results of the investigated set of SDB models and their improvements done with respect to the same processed measurement dataset, it may be concluded that:

- Simple reimplementation of the algorithm using different libraries (i.e., Keras) does not necessarily lead to better results.

- The replacement of linear regression approach with application of neural networks has yielded significant improvement of the algorithm performance. At the same time, the modifications in the algorithm structure like changing the output layer from linear to exponential might worsen the performance.

- Application of regression tree algorithms to the problem of seabed depth estimation presents itself as the best choice in this study. The advantage of retaining the least errors overshadows the problem of results' slight quantification. At the same time, it was demonstrated that the use of more complex random forests' algorithms is not necessary as it does not provide an improvement of performance.

- Significant attention needs to be taken when selecting the inputs for the algorithms. Using the quotient of B2 and B3 reflectances as per reference work for neural network and decision tree implementation was demonstrated as a valid approach. Having that said, it has to be noted that the use of raw reflectance data as inputs performs even better. This is caused by the fact that the former input selection is derived from raw reflectance data. At the same time, the combination of the quotient of logarithms with the raw reflectance inputs proves to be the best approach.

- To sum up, in the presented study the best performing approach in predicting seabed depth proved to be the regression tree algorithm that incorporated as inputs information about

data localization, raw reflectance data from four Sentinel 2 bands, and a quotient of logarithms of B2 and B3 bands.

## Author Contributions

**Conceptualization:** Andrzej Chybicki, Paweł Sosnowski, Tomasz Bieliński, Waldemar Korlub.

**Data curation:** Marek Kulawiak.

**Formal analysis:** Paweł Sosnowski, Tomasz Bieliński.

**Funding acquisition:** Jarosław Parzuchowski.

**Investigation:** Paweł Sosnowski, Tomasz Bieliński, Magdalena Kempa.

**Methodology:** Andrzej Chybicki, Paweł Sosnowski, Tomasz Bieliński, Magdalena Kempa.

**Project administration:** Andrzej Chybicki, Jarosław Parzuchowski.

**Resources:** Tomasz Bieliński, Waldemar Korlub.

**Software:** Paweł Sosnowski, Marek Kulawiak, Tomasz Bieliński, Waldemar Korlub.

**Supervision:** Andrzej Chybicki.

**Validation:** Paweł Sosnowski, Tomasz Bieliński, Zbigniew Łubniewski.

**Visualization:** Paweł Sosnowski.

**Writing – original draft:** Paweł Sosnowski, Magdalena Kempa.

**Writing – review & editing:** Andrzej Chybicki, Zbigniew Łubniewski.

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
