## [Decision Letter · Decision Letter 0]

6 Sep 2022

PONE-D-22-20317Study of Various Machine Learning Approaches for Sentinel-2 Derived BathymetryPLOS ONE

Dear Dr. Chybicki,

Thank you for submitting your manuscript to PLOS ONE. After careful consideration, we feel that it has merit but does not fully meet PLOS ONE’s publication criteria as it currently stands. Therefore, we invite you to submit a revised version of the manuscript that addresses the points raised during the review process.

We look forward to receiving your revised manuscript.

Kind regards,

Emmanuel S. Boss

Academic Editor

PLOS ONE

Journal Requirements:

2.We note that the grant information you provided in the ‘Funding Information’ and ‘Financial Disclosure’ sections do not match. 

3. We note that Figures 3 and 4 in your submission contain [map/satellite] images which may be copyrighted. All PLOS content is published under the Creative Commons Attribution License (CC BY 4.0), which means that the manuscript, images, and Supporting Information files will be freely available online, and any third party is permitted to access, download, copy, distribute, and use these materials in any way, even commercially, with proper attribution. For these reasons, we cannot publish previously copyrighted maps or satellite images created using proprietary data, such as Google software (Google Maps, Street View, and Earth). For more information, see our copyright guidelines: http://journals.plos.org/plosone/s/licenses-and-copyright.

a. You may seek permission from the original copyright holder of Figures 3 and 4 to publish the content specifically under the CC BY 4.0 license.  

Natural Earth (public domain): http://www.naturalearthdata.co

Additional Editor Comments:

Dear authors,

Both reviewers see significant value in your work but have very significant issues with your manuscript presentation as well as analysis.

Please read them carefully as they both spent significant time and effort providing you constructive criticism on how to improve the manuscript.

In particular, both commented on problems with citations (missing, as well as formatting issues) as well as separation between what should be in the Results section (and associated statistics) vs. what should be in the Discussion section.

All the best, Emmanuel

Reviewers' comments:

Reviewer's Responses to Questions

**Comments to the Author**

1. Is the manuscript technically sound, and do the data support the conclusions?

Reviewer #1: Partly

Reviewer #2: Yes

2. Has the statistical analysis been performed appropriately and rigorously? 

Reviewer #1: Yes

Reviewer #2: No

3. Have the authors made all data underlying the findings in their manuscript fully available?

Reviewer #1: No

Reviewer #2: Yes

4. Is the manuscript presented in an intelligible fashion and written in standard English?

Reviewer #1: No

Reviewer #2: Yes

5. Review Comments to the Author

Reviewer #1: Literature is not correct. Some references are missing (i.e., Marks) and some others have wrong numbering.

Line 105, pg 9 of pdf. SDB is not something new. It start back in 80s and already has become an important tool (https://ihr.iho.int/articles/thirty-years-of-satellite-derived-bathymetry-the-charting-tool-that-hydrographers-can-no-longer-ignore/)

In table 1 please use same level of decadal numbers for all values. Also since readers might be confused by the state of both R and R squared, in might worth to use MAE metric and remove R (or provide explanations on what R gives in comparison to R squared).

The message take home at that work is unclear. The results and discussion sectors need to be rewritten and present in a logical approach where the reader can be benefit from reading instead of confusing.

In the methods approach, several band sets have varios data, from the bands per se to the quotient of bands' logarithms etc. Any multicollinearity test has been done to see if there is any redunduncy information?

For the Open Science, please provide a jupiter notebook with the developed code for the reproducability of the work.

Please add notes on what is needed by the user in order to test it with my own data

Reviewer #2: The manuscript is thorough and does investigate an interesting question of what current methods provide the most promising and accurate estimation of SDB. The methods are well laid out and it is clear that a considerable amount of original work went into this study. However, at present, this work is presented more like a report or a white paper rather than a scientific article. The objectives and rationale are poorly defined so while the results are interesting it is unclear why this work has been done and what gaps in our existing knowledge we are trying to fill. I encourage the authors to think carefully about what they are trying to add to the existing body of knowledge.

I also have some specific comments:

The introduction is confusing. It definitely needs more citations and it is unclear what argument you are trying to build. You should identify gaps in the scientific knowledge that you are specifically trying to address and why the gaps are a problem. Having an introduction and a background and theory section is also confusing - these sections should be merged into a more coherent overview and the weaknesses and gaps should be more clearly defined. It is important that the reader gets a firm idea of what problem is trying to be solved and what this paper will contribute to scientific knowledge. Some paragraphs are very short and do not contribute to the body of work convincingly. For example the last two paragraphs of introduction to SDB.

Some references are name and number some are number only

Try to avoid vague sentences such as “In the last years” on line 106. These have little specific meaning.

The introduction to GWR is set out like a list rather than a cohesive argument. It is informative but should be shortened.

The whole introduction sections should be reduced and combined into a single cohesive argument

Lines 282 and 286 - very short paragraphs. Should be rewritten

The different models are well outlined and presented

The results are not thorough. There is no text in the result section and the statistics are very limited. Relying on RMSE and R2 alone is not sufficient. You should also look at %RMSE and standard error. An R2 and RMSE gives you a limited understanding of how well the models performed.

The results are in discussion section but it would be preferable to describe these in the results section. The discussion should discuss these results and place them in context of the existing literature and current body of knowledge. Here you need to confirm that this work has advanced the field. The discussion should also discuss why you are getting the results you have - what drives the performance and error

6. PLOS authors have the option to publish the peer review history of their article (what does this mean?). If published, this will include your full peer review and any attached files.

Reviewer #1: **Yes: **Dimitris Poursanidis

Reviewer #2: No

---

## [Author Response · Author response to Decision Letter 0]

2 Feb 2023

Dear Editor and Reviewers,

Following the latest review of the paper, we would like to resubmit the results of our studies related to the evaluation of Machine Learning approaches to estimate satellite-derived bathymetry.

According to Editor's and Reviewers’ suggestions sent to us on 6th September 2022 (Decision: Revision required [PONE-D-22-20317] ), we have improved the manuscript by following a list of modifications and improvements. 

1) Reproducibility of the experiments and data availability

As to the suggestions raised in the last decision we would like to confirm that all experiments data and scripts that allow reproducing results are available on GitHub and Zenodo platforms. 

In particular, we published bathymetry-estimator code on GitHub:

https://github.com/coast-mapper/bathymetry-estimator

and on Zenodo:

https://zenodo.org/record/6779671

This code can be run with the dataset, which is also published on Zenodo:

https://zenodo.org/record/6543997

These resources are also publicly available and have official DOI assigned: 10.5281/zenodo.6779671 and DOI: 10.5281/zenodo.6543997 respectively.

Instruction on how to run computations with the mentioned dataset can be found in Readme file on GitHub (and inside zip file from Zenodo):

https://github.com/coast-mapper/bathymetry-estimator/blob/main/README.md

Please note, that at the end of the Readme file, Manuscript Reviewers and potential Readers will find commands, which allow them to reproduce scientific experiments as performed in the paper. Near every command in square brackets is the model’s name. The same names were used in the study.

We would like to also add, that due to the complexity of the problem, the technologies used in the research, and the size of the data, we couldn’t share it using a simple ZIP file format. 

2) Structure of the paper:

Due to the Reviewers’ remarks indicating we have made the following improvements to the new version of the manuscript:

- Missing references and numbering have been corrected. 

- We have modified Table 1, namely we removed R (correlation) column since R2 sufficiently describes the results. We also use the same level of decimal numbers. 

- The "Results" and "Discussion" sections have been combined, and their contents, structure and layout have been corrected to make the resulting conclusions more visible. 

According to specific remarks of Reviewer #2 (together with the "specific comments"), the manuscript, and especially the "Introduction" section have been corrected in order to clearly describe the objective and rationale of the research. The following improvements have been made:

- The structure of the "Introduction" has been changed, with adding "The research objectives" subsection at its end. This new subsection contains the authors' explanation precising what the adding to the current state of the art the contents of this paper constitute.

- The SDB and GWR descriptions have been shortened to some extent, but not too much, as in the authors' opinion this content may be interesting for the reader.

- The description of the used measurement data (lines 90 - 97 in the original manuscript) has been moved to the "Materials and methods" section.

- Very short paragraphs have been removed (from the entire manuscript).

- The number of references and citations has been maintained on more-less the same level. As the Introduction section has been shortened, and at the same time the goal and the objectives of the work have been more clearly drawn, in the authors' opinion the number of references and citations is sufficient.

-We have added an additional comment in lines 429-437 (see track changes file) of the revised version of the manuscript regarding the multicollinearity of the data. 

-We have also improved text by minor language corrections, particularly in lines 282-286 of the previous version of the manuscript (short paragraph)

As to additional comments of Reviewer #2 regarding limitations to the statistics, we would like to emphasize that we agree with the Reviewer on a general basis. However, in lines 517-525 of the manuscript, we have justified why we didn’t decide to make this sort of analysis, particularly in this research. The main reason for our decision was that we have already performed such analysis in our previous research:

Mapping south baltic near-shore bathymetry using Sentinel-2 observations, A Chybicki - Polish Maritime Research, 2017

We also think that the graphs presented in Fig. 2 generally show the RMSE error characterization in the depth function, although we agree that the manuscript itself is not discussed on this topic. It also seems to us that a possible additional analysis in this matter would slightly blur the main goal of this study, which is to compare different approaches, and not to focus on a very in-depth analysis of the results for individual algorithms. We also agree with the reviewer on the fact that such an analysis makes the most sense and could be an object of further research in this direction.

All these changes are visible in the track changes MS Word File attached to this correspondence (StudySDB_resubmission_TrackChanges_10_2022.docx)

We hope that the improved manuscript fulfills PLOS ONE requirement for research articles and will be a valuable contribution to the scientific community and PONE Journal Readers. 

Sincerely,

Andrzej Chybicki

---

## [Decision Letter · Decision Letter 1]

4 Sep 2023

Study of Various Machine Learning Approaches for Sentinel-2 Derived Bathymetry

PONE-D-22-20317R1

Dear Dr. Chybicki,

We’re pleased to inform you that your manuscript has been judged scientifically suitable for publication and will be formally accepted for publication once it meets all outstanding technical requirements.

Kind regards,

Bhogendra Mishra

Academic Editor

PLOS ONE

Additional Editor Comments (optional):

Following a careful review of the feedback provided by the reviewers, it is my recommendation that the authors engage in a thorough revision of the manuscript to address concerns related to its structure, language, and analysis. 

Reviewers' comments:

Reviewer's Responses to Questions

**Comments to the Author**

1. If the authors have adequately addressed your comments raised in a previous round of review and you feel that this manuscript is now acceptable for publication, you may indicate that here to bypass the “Comments to the Author” section, enter your conflict of interest statement in the “Confidential to Editor” section, and submit your "Accept" recommendation.

Reviewer #1: All comments have been addressed

Reviewer #3: All comments have been addressed

2. Is the manuscript technically sound, and do the data support the conclusions?

Reviewer #1: Yes

Reviewer #3: Yes

3. Has the statistical analysis been performed appropriately and rigorously? 

Reviewer #1: Yes

Reviewer #3: Yes

4. Have the authors made all data underlying the findings in their manuscript fully available?

Reviewer #1: Yes

Reviewer #3: Yes

5. Is the manuscript presented in an intelligible fashion and written in standard English?

Reviewer #1: Yes

Reviewer #3: Yes

6. Review Comments to the Author

Reviewer #1: Dear Authors

The manuscript has been improved based on the previous suggestions.

The provision of the developed code as docker is good for the reuse by any interested.

Next steo could be a GUI for real easy use.

Best regards

D.

Reviewer #3: The authors considered all comments.

7. PLOS authors have the option to publish the peer review history of their article (what does this mean?). If published, this will include your full peer review and any attached files.

Reviewer #1: **Yes: **Dimitris Poursanidis

Reviewer #3: No

---

## [Editor Report · Acceptance letter]

7 Sep 2023

PONE-D-22-20317R1 

Study of various machine learning approaches for Sentinel-2 derived bathymetry 

Dear Dr. Chybicki:

I'm pleased to inform you that your manuscript has been deemed suitable for publication in PLOS ONE. Congratulations! Your manuscript is now with our production department. 

Kind regards, 

on behalf of

Dr Bhogendra Mishra 

Academic Editor

PLOS ONE